# Comparison between Single and Multi-LED Emitters for Photodynamic Therapy: An In Vitro Study on *Enterococcus faecalis* and Human Gingival Fibroblasts

**DOI:** 10.3390/ijerph19053048

**Published:** 2022-03-05

**Authors:** Morena Petrini, Tania Vanessa Pierfelice, Emira D’Amico, Teocrito Carlesi, Giovanna Iezzi, Camillo D’Arcangelo, Silvia Di Lodovico, Adriano Piattelli, Simonetta D’Ercole

**Affiliations:** 1Department of Medical, Oral and Biotechnological Sciences, University G. d’Annunzio of Chieti-Pescara, Via dei Vestini 31, 66013 Chieti, Italy; morena.petrini@unich.it (M.P.); tania.pierfelice@unich.it (T.V.P.); emira.damico@unich.it (E.D.); teocrito.carlesi@unich.it (T.C.); gio.iezzi@unich.it (G.I.); camillo.darcangelo@unich.it (C.D.); simonetta.dercole@unich.it (S.D.); 2Department of Pharmacy, University “G. d’Annunzio”, Chieti-Pescara, Via dei Vestini, 31, 66100 Chieti, Italy; silvia.dilodovico@unich.it; 3School of Dentistry, Saint Camillus International University of Health and Medical Sciences, Via di Sant’Alessandro 8, 00131 Rome, Italy; 4Dental School, University of Belgrade, 11000 Belgrade, Serbia; 5Fondazione Villa Serena per la Ricerca, 65013 Città Sant’Angelo, Italy; 6Casa di Cura Villa Serena del Dott. L. Petruzzi, 65013 Citta Sant’Angelo, Italy

**Keywords:** photodynamic therapy, LED devices, *Enterococcus faecalis*, human gingival fibroblast

## Abstract

Aim of the study: The aim was to evaluate the effects of two LED devices, TL-01 and TL-03 in photodynamic therapy (PDT), on *Enterococcus faecalis* and on human gingival fibroblasts (HGFs). TL-01, characterized by a single emitter, irradiates one periodontal site at a time, whereas the multi-led device (TL-03) irradiates all vestibular sites of a single arch at a time. Methods: *E. faecalis* bacterial suspensions and HGFs were incubated for 45 min with Aladent gel (ALAD) containing 5-aminolevulinic acid and then exposed to LED devices (ALAD-PDT), having different distance and timing of irradiation (TL-01 N (0.5 mm, for 7 min), TL-03 N (0.5 mm, 15 min) and TL-03 F (30.0 mm, 15 min)). For bacterial suspension, the colony forming units and the live/dead staining were evaluated after 24 h, while the protoporphyrin IX (PpIX) content was monitored in all phases of the experimentation. For HGFs, the cell viability, proliferation, cell morphology, and adhesion were evaluated at 24 h. Results: Both TL-01 and TL-03 showed a significant reduction of bacterial load. The photoinactivation was inversely proportional to the PpIX accumulation. TL-01 and TL-03 promoted proliferation and adhesion of HGFs. Conclusions: Both tested devices for ALAD-PDT were equally effective in significantly reducing *Enterococcus faecalis* growth and in promoting HGFs proliferation and adhesion, in vitro.

## 1. Introduction

Periodontitis and peri-implantitis are two chronic inflammatory pathologies that promote the irreversible break down of the bone and soft tissues that surround teeth and implants. Common etiology of both diseases is the bacterial dysbiosis that promotes tissue destruction, directly because of the presence of proteolytic bacteria and indirectly by triggering the host’s inflammatory response. In both diseases, the chances to treat the contaminated sites successfully are increased with an early diagnosis and intervention. Currently, all available treatments are based on mechanical, chemical, and physical methods that aim to disrupt the bacterial biofilm, promoting the root/implant decontamination [1,2]. Even though there are a wide range of treatments described in the literature, these two pathologies still represent a very tough challenge for clinicians. Moreover, the increase of the problem of antibiotic resistance promoted the dissemination of resistant bacteria and the spreading of iatrogenic infections [3]. In the recent years, the use of light-devices, such as laser and light emitting diodes (LEDs) showed encouraging results against different types of bacteria and yeasts [2,4,5,6,7]. The photoinactivation can be induced by the light activation of endogenous photosensitizers present inside the bacterial cells or by exogenous substances applied by the clinicians, such as in photodynamic therapy (PDT). As suggested by Meimandi et al., PDT is an efficient adjunctive treatment for periodontal disease that, contrarily to antibiotics, is safe and lacks side effects [8]. Another advantage of PDT for the treatment of periodontal disease is the ability to promote a reduction of the inflammatory mediators IL-1β and IL-17 [7]. The photosensitizers used for PDT can be of different nature, such as methylene blue, toluidine blue, and aluminum-chloride-phthalocyanine. However, the use of these substances has been limited due to the staining effect of these dyes post-treatment, which have reduced its overall acceptance [2]. Among the photosensitizers, the delta-aminolevulinic acid within the biosynthetic pathway of heme is the precursor of protoporphyrin IX (PpIX). In turn, PpIX is a specific photosensitizer that, after irradiation with red light (˜630 nm ± 10 nm), induces the production of reactive oxygen species (ROS), provoking directly and indirectly cytotoxic effects in cancer cells and in microbes. The temporary PpIX accumulation observed in bacteria seems to be related to their rates of metabolic activity, which is higher than in mammalian healthy cells. The photokilling occurs through different mechanisms, including photodamage to mitochondria, nuclei, and cell membranes [9,10]. The photodynamic therapy provided by a novel gel containing 5% 5-delta-aminolevulinic acid, Aladent (ALAD) (Alphastrumenti Srl, Melzo, Italy), followed by red LED irradiation (ALAD-PDT), has shown a significant antibacterial action, both against gram-positive and -negative bacteria. In addition, in our recent studies, the antibacterial action against the gram-negative bacteria was significantly higher than the outcome found by authors that used other compounds based on aminolevulinic acid. We supposed that this is due to a synergic effect provided by ROS production because of PpIX accumulation and the presence of specific preservatives in ALAD, such as potassium sorbate and sodium benzoate [11,12]. Another important factor seems to be the pH value, as shown by Greco et al. in a study on *Candida albicans*, in which the antifungal activity of ALAD-PDT is also affected by the acidic pH 3.50 of the ALAD gel [13]. The light source device also represents a crucial tool in the photodynamic therapy, and both Radunovic et al. and Greco et al. irradiated the specimens with an AlGaAs power LED device (TL-01) constituted by 1 LED with 6.00 mm diameter at the exit and a surface irradiance of 380 mW/cm^2^ [11,13]. The use of TL-01, however, could be time-consuming in patients with a great number of periodontal or peri-implant sites to be-treated, because it needs 7 min of irradiation for each site. In order to solve this problem, a novel light emitter, TL-03, specific for the full mouth irradiation and characterized by 32 LED, has been developed and tested. The LED of TL-03 are characterized by a lower irradiance than TL-01, and for this reason, the manufacturer, based on preliminary testing (data not shown), suggested to apply 15 min of irradiation to activate the ALAD gel. TL-01 must applied for 7 min for each periodontal site, while the use of TL-03 could permit to save time in all cases with at least > 3 periodontal pockets. The advantage of TL-03 is that it allows to irradiate all vestibular sites of the single arch simultaneously, saving more time in patients that have multiple periodontal sites to treat. However, contrarily to TL-01, in which the single emitter is specially positioned as close as possible to the site to be treated, in the event that several sites are irradiated simultaneously with TL-03, it is possible that the irradiation distance may increase. Thus, before proceeding with clinical trials, in vitro studies are required to verify the effectiveness of this new device. Therefore, the first purpose of this study was to compare the antibacterial effect of ALAD-PDT on *Enterococcus faecalis* and the cytocompatibility on human gingival fibroblasts (HGFs), irradiated with the traditional device (TL-01) at 0.5 mm of distance that already showed antimicrobial and antifungal inactivation [11,12,13], and with the new device TL-03. The secondary purpose was to compare the ALAD irradiated with TL-03 at two different distances of irradiation: TL-03 N near (d = 0.5 mm) and TL-03 F far (d = 30.0 mm).

## 2. Materials and Methods

### 2.1. Light Sources

TL-01: constituted by 1 LED with 6 mm diameter and a surface irradiance of 380 mJ/cm^2^ (ALPHA Strumenti Srl, Melzo, MI, Italy) (Figure 1A).

TL-03: the handpiece is characterized by a multiple 32× Power LED with a surface irradiance of 6 mJ/cm^2^ (ALPHA Strumenti Srl, Melzo, MI, Italy) (Figure 1B).

### 2.2. Experimental Design

The experiments were performed using *Enterococcus faecalis* and human gingival fibroblasts incubated with ALAD gel to evaluate the effects of three test groups based on two LED devices, irradiance, distance, and timing of irradiation:TL-01 N (near) = 0.5 mm distance between the samples and the light source, 7 min of irradiation (Figure 1C).TL-03 N (near) = 0.5 mm distance between the samples and the light source, 15 min of irradiation (Figure 1D).TL-03 F (far) = 30.0 mm distance between the samples and the light source, 15 min of irradiation (Figure 1E).

The untreated and unexposed cells are considered the control (CTRL).

All test groups were incubated for 45 min with 100% *v*/*v* Aladent gel containing 5% of delta-aminolevulinic acid (5-ALA), labeled as ALADENT (ALAD), commercialized by ALPHA Strumenti Srl (Melzo, MI, Italy), and subsequently irradiated with the two devices. In all the experiments, the devices were mounted perpendicularly to the wells using a particular polystyrene box to maintain a constant distance of irradiation, as indicated above. The irradiation was performed under a laminar flow hood in the dark under aseptic conditions. Upon incubation and irradiation, *E. faecalis* was tested for the following experiments: (i) colony forming units; (ii) live/dead assay; (iii) intracellular detection of PpIX. After the incubation and irradiation, HGFs were assessed for the following experiments: (i) cell viability and proliferation assay; (ii) cell adhesion and morphology. All analyses were performed three times in triplicate.

### 2.3. Strain Culture Condition

*Enterococcus faecalis* ATCC 29212 bacterial suspension was grown overnight at 37 °C in brain heart infusion (BHI) broth and prepared at the concentration to 9.270 log10 colony forming units, log10(CFU/mL), using a spectrophotometer (Agilent Technologies 8453 UV, Santa Clara, California USA). Aliquots of 1 mL were dispensed in triplicate into 25-well (dimension: 20 mm ∙ 20 mm) flat-bottom plates for each treatment group.

### 2.4. Colony Forming Units (CFU) Determination

At the end of each treatment, selected 10-fold dilutions of the samples, including CTRL, were plated on MacConkey plates without crystal violet (DifcoTM, Becton, Dickinson, and Company; Sparks, NV, USA) and incubated overnight al 37 °C. After 24 h, the number of colony forming units per milliliter (CFU/mL) was determined.

### 2.5. Live/Dead Assay

The *E. faecalis* cell viability was evaluated by a fluorescent microscopy using a BacLight LIVE/DEAD Viability Kit (Molecular Probes, Invitrogen detection technologies, USA) as previously described [5,6]. After treatments, 1 mL of the bacterial suspension was concentrated by centrifugation at 10,000× *g* for 10–15 min, washed with PBS, and stained as indicated by the manufacturer. After 15 min, 10 μL of the stained bacterial suspension was posed on a slide, and an 18 mm square coverslip was put on top by adhering to the suspension. The images were observed at fluorescent Leica 4000 DM microscopy (Leica Microsystems, Milan, Italy). The observations were recorded at an emission wavelength of 500 nm for SYTO 9 that stains viable cells with a green, fluorescent signal and of 635 nm for propidium iodide and cells with impaired membrane activity with a red fluorescent signal.

### 2.6. Intracellular Detection of PpIX

To investigate the time-dependent intracellular accumulation of PpIX, 200 µL of *E. faecalis* ATCC 29212 suspension was dispensed in triplicate into 96-well plates for each test group and for the control. Four different test groups were distinguished: (i) ALAD gel treatment without any incubation time (ALAD t0); (ii) ALAD gel treatment incubated for 45 min (ALAD t45); (iii) ALAD gel treatment incubated for 45 min and LED TL01 irradiated for 15 min (TL-01); (iv) ALAD gel treatment incubated for 45 min and LED TL03 irradiated for 15 min (TL-03). *E. faecalis* suspension without any active substance or irradiation was used as the control (CTRL). To extract PpIX from bacteria, 100 µL of the 0.5 M HClO4 in 50% methanol solution was added [14]. PpIX fluorescence was then measured at 608 nm after excitation with 405 nm, using a microplate spectrofluorometer (Synergy H1 Hybrid BioTek Instruments, Winooski, VT, USA).

### 2.7. Viability and Proliferation Assay

Human gingival fibroblasts (HGFs) from ATCC (Manassas, Virginia, USA) were cultured in DMEM low glucose (Corning, New York, New York, USA) supplemented with 10% fetal bovine serum (FBS, Gibco-Life Technologies, Monza, Italy), 1% penicillin, and streptomycin at 37 °C in a humidified atmosphere of 5% CO_2_. HGF cells were seeded into 96-well/plates at a density of 6·10^3^ cells/well. After 24 h of culture, cells were incubated to 100% ALADENT (100-ALAD) gel in serum-free medium for 45 min, as suggested by the manufacturer. The cultures were then exposed to an irradiation by LED devices (TL-01, TL-03) for 15 min. Subsequently, the cells were washed thrice with PBS and cultured in medium containing 10% fetal bovine serum (FBS), and the effects of photodynamic therapy were assessed up to 24 h after treatments. At one day post treatments, 10 µL/well of the MTS solution (CellTiter-96 Promega, Madison, WI, USA) was added at 37 °C for 1 h. The absorbance was measured using a microplate reader (Synergy H1 Hybrid BioTek Instruments, Winooski, VT, USA) at 490 nm. The results were expressed in the form of percentage comparing each group test to the control.

### 2.8. Morphological Characterization of Adherent Cells

The evaluation of human fibroblast morphology was analyzed by SEM. An amount of 1 × 10^4^ cells were cultivated onto the surfaces of glass cover slips (Thermo Fisher, Waltham, MA, USA) into the 24-well plate. In brief, gingival fibroblasts were subjected to ALAD-PDT, and after 24 h, adhered cells to the cover slips were fixed with 4% glutaraldehyde solution and washed with PBS 0.1 M. The cells were treated with Osmium Tetroxide (1% in PBS) for 1 h and rinsed with ultra-high quality (UHQ) water. Subsequently, the samples were dehydrated with increasing alcohol concentrations (50, 75, 95, and 100%) for 15 min each, followed by hexamethyldisilazane (HMDS, Sigma-Aldrich, St. Louis, MO, USA) overnight. The cover slips were then mounted onto aluminum stubs and gold-coated in an DSR1Desk Sputter Coater/Carbon Evaporator (Nanostructured Coatings Co., Tehran, Iran) before imaging by means of SEM (Phenom-World BV, Eindhoven, The Netherlands). Images were taken using an accelerating voltage of 15 kV with the backscattered electronic signal detector (BSE), BSD full, to obtain images of cells of a different color (white) than the cover slip (gray).

### 2.9. Statistical Analysis

Statistical analysis was performed using GraphPad Prism 8 (GraphPad Software, San Diego, CA, USA), and to compare two groups, a T test was used for viability assay.

For other variables, the Levene test permitted the evaluation of the homogeneity; in the case of homogeneous groups, the ANOVA and post-hoc Tukey test was adopted. In the case of heterogeneous variances, outliers were limited, and then the Levene test was repeated for confirming the homogeneity among variables.

A *p*-value < 0.05 was considered significant.

## 3. Results

### 3.1. Colony Forming Units (CFUs) Determination

The effects of photodynamic therapy on *E. faecalis* colony forming units are shown in Figure 2. There was a significant photoinactivation in all exposed groups with respect to the unexposed control (* *p* < 0.001). No significant differences were found between TL-01 N vs. TL-03 N or between TL-03 N vs. TL-03 F. On the contrary, a significantly higher antibacterial effect was found in TL-01 N vs. TL-03 F (** *p* = 0.025).

### 3.2. Live/Dead Assay

Images of live/dead staining of *E. faecalis* bacterial suspension after un-exposition and exposition to ALAD-PDT based on two different devices at different distances are represented in Figure 3.

The live/dead assay revealed analogous effects for TL-01 N and TL-03 N, with respect to the control, while TL-03 F irradiation induced a slightly lower percentage of cell death. No statistically significant difference was recorded among test groups TL-01 N, TL-03 N, and TL-03 F.

### 3.3. Intracellular Detection of PpIX

The level of intracellular PpIX, after 5-ALAD addition, was fluorometrically detected in *E. faecalis* and shown in Figure 4. The highest PpIX content resulted immediately after the addition of ALAD gel without any irradiation, and it remained high after 45 min of incubation, even though it slightly diminished. On the contrary, the PpIX fluorescence returned to the same level of the control when the incubation with ALAD gel was followed by LED light irradiation. The measurement also revealed any significant difference between the two LED devices, TL-01 and TL-03.

### 3.4. Viability and Proliferation Assay

The results of the MTS assay, shown in Figure 5, indicate that both LED devices TL-01 and TL-03 in combination with ALAD gel exerted similar effects on gingival fibroblasts, regardless of the distance of the light source (near or far). In addition to the absence of the cytotoxicity, all treatments significantly promoted cell proliferation compared to untreated and unexposed cells (CTRL).

### 3.5. Cellular Morphology Characterization

The images from SEM analysis of cell adhesion and morphology are shown in Figure 6. The classical fibroblast spread morphology was observed. TL-01 and TL-03, as light source devices in the photodynamic therapy in combination with ALAD gel, both enhanced cell adhesion compared to the control, which was the untreated and unexposed cells in a similar manner. In the TL-01 group, cellular extension in the form of filopodia and cell-to-cell communication network can be observed. Typical, polygonal, spindle-shaped HGFs attached, and the spread was shown in TL-03 group.

## 4. Discussion

Current treatments of periodontitis and peri-implantitis have the main objective of bacterial and biofilm removal by means of chemical/physical disinfection and mechanical decontamination of periodontal and peri-implant sites. However, less importance is given to the effect of these treatments on the surrounding tissues, considering as a goal the achievement of the lack of cytotoxicity. In this study, the effects of ALAD-PDT based on the two different LED devices were compared on *Enterococcus faecalis* and HGFs. The traditional device TL-01, with an irradiation distance of 0.5 mm, which already showed antimicrobial and antifungal inactivation [11,12,13], was compared with another device, TL-03, specifically manufactured for full-mouth application. TL-03 could reduce the time of treatment, but, contrarily to TL-01, the distance of application of the light could not always be as close as possible. The use of a specific light wavelength, distance, and energy are fundamental for the success of PDT in clinical practice. Besides the range of light and the energy content, the delivery of light for PDT is characterized by the size, shape, and anatomical location of the site to treat [15]. Furthermore, the intensity of the produced light should be uniform to allow for dose calculations during treatment. In this context, we tested TL-03 at two different distances: 0.5 mm (TL-03 N), and as far away as possible, 30.00 mm (TL-03 F). In our study, the colony forming units of all three groups exposed to ALAD-PDT significantly decreased with respect to control. No differences were found between the two devices TL-01 N and TL-03 N, and no significant differences were revealed between TL-03 N and TL-03 F. However, the photoinactivation was higher with a lower distance of irradiation (N), and a significant difference was found between TL-01 N and TL-03 F. The live/dead staining confirmed the results of the CFUs experiment: all groups exposed to ALAD-PDT showed a significant higher proportion of dead cells, with respect to the control. The obtained results are very interesting with significant clinical implications, although there is no comparation with a positive control. In addition, these results are in accordance with Radunovic et al. and Petrini et al., who showed the effectiveness of ALAD-PDT against gram-positive and -negative bacteria [11,12]. Although the effect of irradiation distance was previously studied in literature, it is very difficult to understand how the irradiance pattern for several distances can exert its influence, because many factors are involved, such as the emitting angle, the optics design, the shape of the LEDs, and the material used for the LED coating [16]. D’Ercole et al. recorded a significant reduction of bacterial photoinactivation by increasing the distance of irradiation from 0.5 to 10.0 mm by using a multi-led device at 880 nm [4]. Although photodynamic therapy has shown efficacy in treating microbial infections, the mechanisms by which this method inactivates microbes are not completely understood. While 5-ALA itself is not photodynamically active, it can act as a prodrug that induces the endogenous accumulation of protoporphyrin inside target cells, and the followed light activation leads to antimicrobial effects. Our results with *E. faecalis* showed that the production of PpIX from exogenously added 5-aminolevulinic acid in ALAD gel is time-dependent. As it is already known, exogenous 5-ALA is taken up by bacteria via specific transport systems, inducing an accumulation of protoporphyrin IX [17] that can be observed in our experiment when the fluorescence of PpIX has been measured after the addition of ALAD gel to *E. faecalis* solution. The subsequent light irradiation of endogenous PpIX would lead to the death of microbes by means of various cellular damages [18,19]. The increased levels of protoporphyrin that was recorded after 45 min of incubation with ALAD clearly diminished upon LED irradiation, regardless of the light source used. This agrees with the PDT antimicrobial strategy that necessarily involved a photosensitizer and a light at specific wavelength. No significant differences were observed between the three test groups, concerning the levels of PpIX after the irradiation time. Beyond the antibacterial activity, any protocol for oral application should exert no cytotoxic effects towards healthy cells. Thus, in this study the cytocompatibility of ALAD-PDT was investigated on human gingival fibroblasts. The results of the viability assay showed how the light-emitting devices both exert proliferative effects on fibroblast cells in a similar manner. In addition, the same pro-adhesive influence was observed with SEM when fibroblasts were subjected either to TL-01 or TL-03. Although to date there are few studies concerning the effects of 5-ALA-based PDT on fibroblast cells, our results are in accordance with two other investigations. Moore et al., in 2005, reported proliferative abilities of red light on dermal fibroblasts during wound healing [20]. Haddad et al. observed that 5-aminolevulinic acid did not negatively affect the viability of murine fibroblasts [21]. Altogether, our results may indicate that one of the advantages of LED-based systems is the shape of diffusers that may be differently made to achieve similar outcomes.

## 5. Conclusions

TL-01 and TL-03 used in association with ALAD gel were equally effective in reducing *Enterococcus faecalis* and in promoting the proliferation and adhesion of HGFs, in vitro. No significant differences were found when increasing the distance of irradiation, between 0.5 and 30.0 mm; however, it is recommended to use the lower distance of irradiation, in order to gain the best results.

## Figures and Tables

**Figure 1 ijerph-19-03048-f001:**
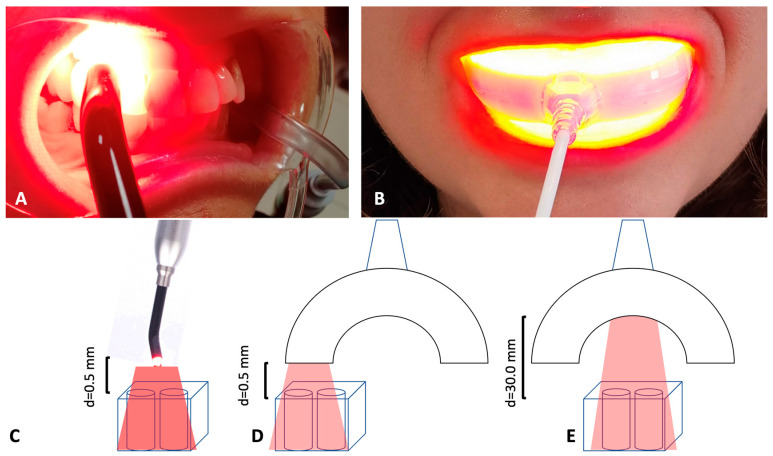
LED devices: (**A**) TL-01, consisting of a single emitter with 6 mm diameter and a surface irradiance of 380 mJ/cm^2^. TL-01 irradiates one periodontal site at a time. (**B**) TL-03: multi-LED device irradiates all vestibular sites of a singular arch. It is characterized by a multiple 32× Power LED with a surface irradiance of 6 mJ/cm^2^. (**C**–**E**) Distances of irradiation between the samples and the light source.

**Figure 2 ijerph-19-03048-f002:**
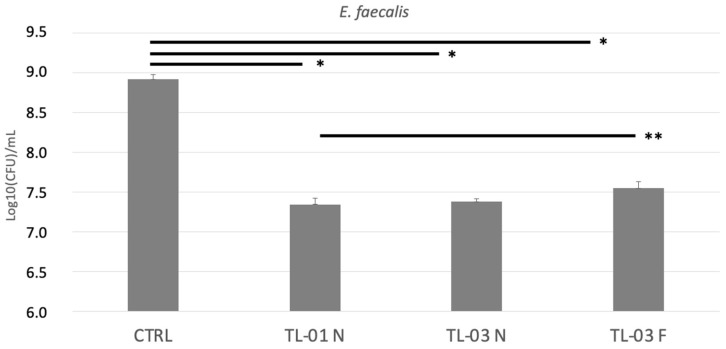
Effects of photodynamic therapy on *E. faecalis* colony forming units (log10CFU/mL) after 24 h. Data are presented as mean ± standard deviation (SD) of three independent experiments in triplicate and are expressed relative to untreated and unexposed bacteria (CTRL). TL-01 N (near); TL-03 N (near); TL-03 F (far). Tukey test showed * *p* < 0.01; ** *p* = 0.025 among the groups.

**Figure 3 ijerph-19-03048-f003:**
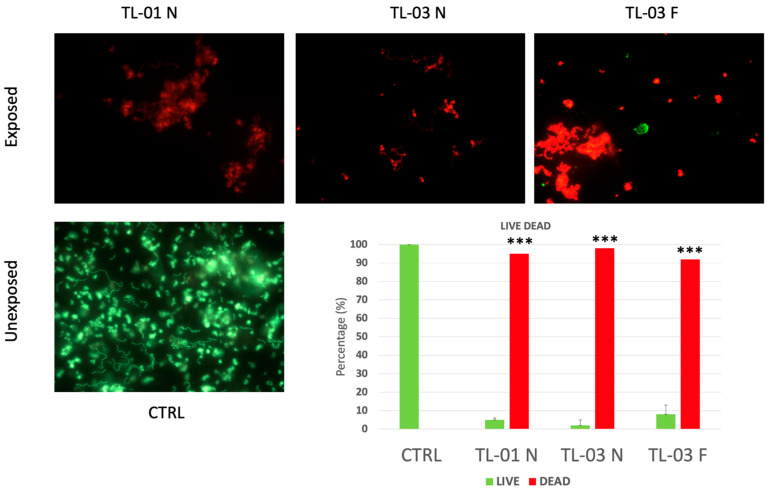
Live/dead staining of *Enterococcus faecalis* growth for 24 h, treated with ALAD, and irradiated with tested devices at different distances: TL-01 N, TL-03 N, and TL-03 F. Histogram shows the percentages of viable (green bars) and dead (red bars) cells. Untreated and unexposed cells were considered as control (CTRL). Tukey test showed *** *p*-value < 0.05 between all groups and CTRL. TL-01 N (near); TL-03 N (near); TL-03 F (far).

**Figure 4 ijerph-19-03048-f004:**
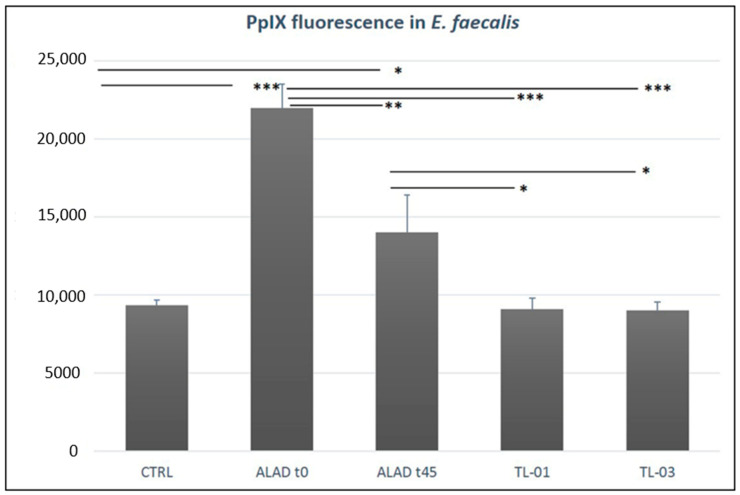
Intracellular accumulation of PpIX. In the absence of irradiation, a significant increase of PpIX fluorescence was observed in *E. faecalis* solution treated with ALAD gel immediately after its addition and after 45 min of incubation. After the irradiation by two tested LED devices, TL-01 and TL-03, the PpIX levels were not enhanced. In addition, there was not any statistical difference between TL-01 and TL-03, nor with respect to the control. Data are presented as mean ± SD of three independent experiments in triplicate. Tukey tests were calculated between the test groups and the control and among test groups (* *p*-value < 0.05; ** *p*-value < 0.01; *** *p*-value < 0.001).

**Figure 5 ijerph-19-03048-f005:**
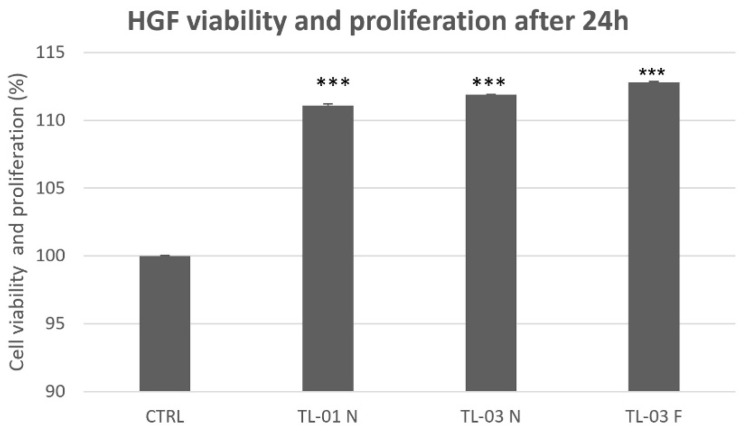
Effects of photodynamic therapy on HGFs viability and proliferation after 24 h. No cytotoxicity effects were observed within all groups. Ttest showed significant differences between each test group and control. (*** *p* < 0.001). TL-01 N (near); TL-03 N (near); TL-03 F (far).

**Figure 6 ijerph-19-03048-f006:**
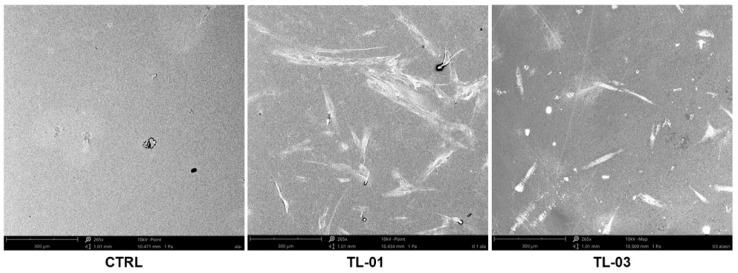
Morphology and adhesion of HGFs at 265× magnification. Images showed the increased fibroblast adhesion after the exposure to the irradiation by light from TL-01 and TL-03, upon ALAD gel treatment. Scale bars: 300 μm. Images were taken using an accelerating voltage of 15 kV.

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
