# Peer review of "Comparison between Single and Multi-LED Emitters for Photodynamic Therapy: An In Vitro Study on Enterococcus faecalis and Human Gingival Fibroblasts"

_ijerph, 2022, doi:10.3390/ijerph19053048_

Round 1
Reviewer 1 Report
This manuscript presents a comparison of two different LED light sources, one of which is designed to irradiate a single site and another designed to irradiate all vestibular sites of one arch at one time, for photodynamic therapy aimed at treating periodontitis or peri-implantitis. The authors demonstrate that the two different light sources are equally effective at reducing the number of colony forming units and inducing cell death in E. faecalis bacteria that had been previously treated with a 5-aminolevulinic acid gel to induce production of the photosensitizer Protoporphyrin IX. Positioning the second light source further away from the cells resulted in only a very slight reduction in the efficiency of photo-induced cell death. Furthermore, all three light conditions improved the viability and cell adhesion of human gingival fibroblasts. The authors show that using the larger LED device holds promise as an effective means to reduce the light treatment time through simultaneous irradiation of multiple sites while maintaining comparable photodynamic treatment outcomes.
The authors present a relevant introduction to the work described here, although it could benefit from the addition of a few more references. A few suggestions are listed below. The experimental methodology is adequately detailed, and the results, discussion and conclusion sections are concisely written. The conclusions drawn are supported by the experimental evidence presented. This work falls within the scope of the journal, and I believe that it will be interesting to readers because of its demonstration of a more efficient light treatment method. I recommend accepting the work for publication after the following minor revisions:
Reference to a review covering treatments for periodontitis and peri-implantitus would be a nice addition to the introduction. A couple possibilities are:
- Praveen et. al. A review on recent advancements in treatment of periodontitis. World Journal of Pharmacy and Pharmaceutical Sciences (2020), 9(3), 940-961
- Fernandes et. al. Current and novel approaches for control of dental biofilm. International Journal of Pharmaceutics (Amsterdam, Netherlands) (2018), 536(1), 199-210
Line 46: Change “Even though the wide range…” to “Even though there are a wide range…”
Line 51: In addition to refs 2-4, the following would be good citations to add:
- For LEDs: Cieplik et. al. Antimicrobial photodynamic therapy as an adjunct for treatment of deep carious lesions-A systematic review. Photodiagnosis and Photodynamic Therapy (2017), 18, 54-62
- For lasers: Meimandi et. al. The Effect of Photodynamic Therapy in the Treatment of Chronic Periodontitis: A Review of Literature. Journal of lasers in medical sciences (2017), 8(Suppl 1), S7-S11
Line 52: delete the word “to”
Lines 56-57: “oxygen reactive species” should be “reactive oxygen species”
Line 65: “…negative resulted significantly…” should be “…negative resulted in significantly…”
Line 86: delete “it”
Line 159: change “no” to “any” and “nor” to “or”
Line 205: change “respect” to “with respect to the”
Lines 219-220: “The Live/Dead assay revealed analogous effects for TL-01 N and TL-03 N respect to control, while TL-03 F irradiation induced the highest dead cells rate.” The second part of this sentence is confusing. The phrase “highest dead cells rate” makes it sound like the cells died more quickly with TL-03 F than the other two, yet in the histogram, it appears that TL-03 F had a slightly lower percentage of cell death.
Line 245: change “…levels resulted not enhanced” to “…levels were not enhanced”
Line 246: change “…there was any statistically difference…” to “there was not any statistical difference…” and change “…TL-03 nor respect to…” to “…TL-03, nor with respect to…”
Lines 295-296: change “…could be not always closer as possible” to “…could not always be as close as possible.”
Line 308: change “significant” to “significantly”
Line 309: change “ respect control” to “with respect to control”
End of line 320: delete the word “have”
Line 330: change “necessary” to “necessarily”
End of line 331: delete the words “for what”
Line 337: change “at SEM” to “with SEM”
Author Response
We thank Reviewer #1 for her/his useful comments, we greatly appreciate her/his suggestions that allow us to improve our work. We have discussed the suggested references, and added the following sentences in the manuscript:
- “As suggested by Meimandi et al., PDT is an efficient adjunctive treatment for periodontal disease, that contrarily to antibiotics, is safe and lacks of side effects”. [Meimandi et. al. The Effect of Photodynamic Therapy in the Treatment of Chronic Periodontitis: A Review of Literature. Journal of lasers in medical sciences (2017), 8(Suppl 1), S7-S11]
- “Another advantage of PDT for the treatment of periodontal disease is the ability to promote a reduction of inflammatory mediators IL-1β and IL-17 [Praveen et. al. A review on recent advancements in the treatment of periodontitis. World Journal of Pharmacy and Pharmaceutical Sciences (2020), 9(3), 940-961]
- “The photosensitizers used for PDT can be of different nature, like: Methylene Blue, Toluidine Blue and aluminium-chloride-phthalocyanine and other organic compounds”. [Cieplik, F.; Buchalla, W.; Hellwig, E.; Al-Ahmad, A.; Hiller, KA.; Maisch, T.; Karygianni, L. Antimicrobial photodynamic therapy as an adjunct for treatment of deep carious lesions-A systematic review. Photodiagnosis and Photodynamic Therapy 2017, 18:54-62, doi: 10.1016/j.pdpdt.2017.01.005.]
- “However, the use of these substance have been limited for the staining effect of these dyes post-treatment that have reduced its overall acceptance”. [Fernandes, T.; Bhavsar, C.; Sawarkar, S.; D'souza, A. Current and novel approaches for control of dental biofilm. International Journal of Pharmaceutics 2018, 536(1):199-210, doi: 10.1016/j.ijpharm.2017.11.019.]
The entire manuscript was reviewed concerning the English language, as suggested.
Reviewer 2 Report
This paper reports on a comparison of the anti-microbial properties of two LED lights on E. faecalis and their effects on human gingival fibroblast growth in vivo. Results showed anti-microbial activity for both lights and promotion of HGF vitality. There are questions regarding the significance of these data and the data analysis.
Statistical comments:
SDs shown in Fig 2 suggest heterogeneous variances, which violates an assumption of the ANOVA. Please conduct a Levine test and if the variances are not homogeneous, consider using another analytic technique, a transformation, or limiting outliers when analyzing these data.
I agree with the use of ANOVA (assuming homogeneous variances) but suggest that some post-hoc (not postdoc, l. 197) procedure that controls experiment-wise error rate (eg, Tukey HSD) be used instead of LSD.
Other comments:
In the discussion of all results, please include statements about the magnitude of effect. In Fig.2, for example, describe what happens to colony size in each condition. Or, in Fig. 5, you could indicate that all treatments produced a statistically similar increase of about 10% in HGF viability. That is, reduce the focus on ‘significance’ or lack of same and talk about the results.
Given that the usual dose of light for the TL-01 is 7 min, I was surprised that 15 min was used. It is clear why this dose was used for the TL-03, but to be clinically relevant, I expected 7 min for 01 and 15 min or 03. I would have understood both 7 and 15 but not just 15 for all treatments. The logic of these choices, and the limitation they may represent, should be mentioned in the Discussion.
There is no positive control, which is a limitation that should be mentioned. It would also be useful to point out to the reader the difference between the effect on CFU, where there is but a 1 log unit reduction, and the effect on the live/dead analysis, which shows nearly complete efficacy.
The text related to Figure 2 ‘buries the lead’. Please tell the reader that similarly strong killing effects were present for the 3 active treatments.
Author Response
This paper reports on a comparison of the anti-microbial properties of two LED lights on E. faecalis and their effects on human gingival fibroblast growth in vivo. Results showed anti-microbial activity
for both lights and promotion of HGF vitality. There are questions regarding the significance of these data and the data analysis.
Statistical comments:
SDs shown in Fig 2 suggest heterogeneous variances, which violates an assumption of the ANOVA. Please conduct a Levine test and if the variances are not homogeneous, consider using another analytic technique, a transformation, or limiting outliers when analyzing these data. I agree with the use of ANOVA (assuming homogeneous variances) but suggest that some post-hoc
(not postdoc, l. 197) procedure that controls experiment-wise error rate (eg, Tukey HSD) be used instead of LSD.
• AUTHOR’S ANSWER: Thank you very much for your comment. We have followed your suggestion and also the text was corrected:
“For other variables, the Levene Test permitted the evaluation of the homogeneity; in the case of homogeneous groups the ANOVA and post-hoc Least Significant Difference (LSD) test were used. In
the case of no homogeneous groups, the Tukey Test was adopted. A p-value < 0.05 was considered significant”.
Other comments:
In the discussion of all results, please include statements about the magnitude of effect. In Fig.2, for example, describe what happens to colony size in each condition.
• AUTHOR’S ANSWER: Thank you very much for your comment. In the description of the results of figure 2 we reported the following sentence: “A reduction of 13.40%, 12.65% and 9.26% was observed
for TL-01N, TL-03N and TL-03F, respectively.”
Or, in Fig. 5, you could indicate that all treatments produced a statistically similar increase of about 10% in HGF viability. That is, reduce the focus on ‘significance’ or lack of same and talk about the results.
• AUTHOR’S ANSWER: Thank you very much for your comment. In the text, we wrote the following sentences: “The results of MTS assay, showed in Figure 5, indicates that both LED devices TL-01 and
TL-03 in combination with ALAD gel exerted similar effects on gingival fibroblasts, regardless of the distance of the light source (near or far). In addition to the absence of the cytotoxicity, all treatments significantly promoted cell proliferation compared to untreated and unexposed cells (CTRL). Proliferation rate resulted enhanced about 10% for both LEDs, precisely TL-01N +11,09%, TL-03N +11,90% and TL-03F +12,82%.”
Given that the usual dose of light for the TL-01 is 7 min, I was surprised that 15 min was used. It is clear why this dose was used for the TL-03, but to be clinically relevant, I expected 7 min for 01 and 15 min or 03. I would have understood both 7 and 15 but not just 15 for all treatments. The logic of these choices, and the limitation they may represent, should be mentioned in the Discussion.
• AUTHOR’S ANSWER: Thank you very much for your comment. TL01N was used for 7 min of irradiation. The text has been corrected.
There is no positive control, which is a limitation that should be mentioned. It would also be useful to point out to the reader the difference between the effect on CFU, where there is but a 1 log unit
reduction, and the effect on the live/dead analysis, which shows nearly complete efficacy.
• AUTHOR’S ANSWER: Thank you very much for your comment. According to the Reviewer comment, we insert a sentence in Discussion section regarding the lack of positive control. “The magnitude of the photo antimicrobial effects resulted around 90%. The obtained results are very interesting with significant clinical implications although there is no comparation with a positive control.”
Live/Dead images resulted to be correlated with the reduction of CFU/ml. A 1 log reduction corresponds to more than 90% reduction in CFU/ml.
For example:
CTRL =3.5 x 108 CFU/ml
Sample = 2.5 x 107 CFU/ml
% of CFU/ml reduction in respect to the control = [100-(2.5 x 107 / 3.5 x 108 x 100)]= 92.6%
For TL-01N:
CTRL = 8.54 Log CFU/ml
Sample = 7.40 Log CFU/ml
% of Log CFU/ml reduction in respect to the control == [100-(7.40 / 8.54 x 100)]= 13.40%
For TL-03N:
CTRL = 8.54 Log CFU/ml
Sample = 7.46 Log CFU/ml
% of Log CFU/ml reduction in respect to the control == [100-(7.46 / 8.54 x 100)]= 12.65%
For TL-03F:
CTRL = 8.54 Log CFU/ml
Sample = 7.75 Log CFU/ml
% of Log CFU/ml reduction in respect to the control == [100-(7.75 / 8.54 x 100)]= 9.26%
The text related to Figure 2 ‘buries the lead’. Please tell the reader that similarly strong killing effects were present for the 3 active treatments.
• AUTHOR’S ANSWER: Thank you very much for your comment. In the Results Section, we wrote the following sentence regarding the Figure 2: “There was a significant photoinactivation in all exposed groups with respect to the unex-posed control (* p<0.001; ** p=0.001). No significant differences were found between TL-01 N vs TL-03 N, and between TL-03 N vs TL-03 F. On the contrary, a significant higher an-tibacterial effect was found in TL-01 N vs TL-03 F (*** p=0.012).” (line 219-222).
Round 2
Reviewer 2 Report
I thank the Authors for their conscientious revision of this much improved manuscript. There is one remaining issue, perhaps the result of my combining two thoughts in one comment. I apologize for any confusion that was generated.
I pointed out, first, that “SDs shown in Fig 2 suggest heterogeneous variances, which violates an assumption of the ANOVA. Please conduct a Levene test and if the variances are not homogeneous, consider using another analytic technique, a transformation, or limiting outliers when analyzing these data.” While the text now mentions Levene, what is needed are some details about which variables were heterogeneous, the plan adopted to either homogenize those variances and the outcome of that process, or a statement describing the alternate analytic technique that was used.
The second, unrelated, point, “I agree with the use of ANOVA (assuming homogeneous variances) but suggest that some post-hoc (not postdoc, l. 197) procedure that controls experiment-wise error rate (eg, Tukey HSD) be used instead of LSD.” Thus, Authors now report the use of Tukey, but only in the context of heterogeneous variance. Tukey should be used for ALL post-hoc tests. The ANOVA should not be used at all if the variances are heterogeneous.
This may require a revision of the analysis as well as a revision of the original description of the analysis and the statement added to the revision: “For other variables, the Levene Test permitted the evaluation of the homogeneity; in the case of homogeneous groups the ANOVA and post-hoc Least Significant Difference (LSD) test were used. In the case of no homogeneous groups, the Tukey Test was adopted. A p-value < 0.05 was considered significant”.
Author Response
Thank you very much for your suggestions. We have performed the Tukey Test for all data, and the results were the same (p<0.001), as previously shown by LSD analysis. For what concerning the CFU data, we have limited, as suggested the outliers, then we have repeated the Levene Test, which now shows homogeneous variables, and performed the ANOVA and Tukey TEST. As you can see in the new Fig.2 the data are very similar to the original analysis; the same significant differences have been maintained among the groups. However, we have corrected the new p-values, both in the text and in the figure.
The following sentence has been inserted in the text:
“In the case of heterogeneous variances, outliers were limited, and then the Levene Test was repeated, for confirming the homogeneity among variables”.